# Obj2CAD: Object-Oriented Text-to-CAD Generation with Large Language Models

## Abstract

Text-to-CAD aims to generate CAD models directly from natural language descriptions. Existing methods predominantly follow a procedural paradigm that represents modeling as long sequences of operations. This approach suffers from several inherent limitations: strictly order-dependent, prone to error accumulation, cluttered with redundant low-level constraints, and misaligned with the object-centric reasoning of both human designers and large language models (LLMs). We introduce **Obj2CAD**, the first framework to shift text-to-CAD from a procedural to an object-oriented paradigm. To support this shift, we curate a new dataset of 1,000 examples, including both industrial parts and carefully selected shapes that are linguistically describable. Each example is converted into an object-oriented representation that emphasizes hierarchical structure and semantic constraints while de-emphasizing redundant low-level constraints. Building on this foundation, we design an LLM-driven framework that combines top-down planning with bottom-up generation, offering a divide-and-conquer approach to text-to-CAD. To enhance spatial reliability, we propose geometric assembly reasoning to formulate assembly explicitly as geometric-mathematical problems. Finally, we introduce an interactive iterative mechanism that incorporates user feedback to refine objects and expand the object graph, enabling continuous system improvement. We provide a **live demo** of Obj2CAD and will publicly release the dataset to support future research on object-oriented CAD generation.

## 1 Introduction

*Text-to-CAD* has emerged as a promising direction to generate computer-aided design (CAD) programs directly from natural language (Zhang et al., 2024; Xie et al., 2025; Khan et al., 2024). This capability lowers the barrier to 3D modeling, automates repetitive design tasks, and enables human-AI collaboration. Beyond its practical value in industrial and product design (Heidari & Iosifidis, 2025), it also connects to broader trend of exploring reasoning and generation capabilities of large language models (LLMs) across modalities (OpenAI, 2023a; Dubey et al., 2024; OpenAI, 2025).

Most existing work relies on datasets such as DeepCAD (Wu et al., 2021) and Fusion 360 Gallery (Willis et al., 2021), which are derived from logged modeling processes of CAD software, such as operations like *sketch*, *extrude*, and *fillet*. While invaluable for advancing research on CAD automation and supporting a wide range of deep learning-based generation methods (Zhang et al., 2024; Xie et al., 2025; Khan et al., 2024), these approaches fundamentally adopt a *procedural* modeling paradigm, where a geometric object is reconstructed through long ordered sequences. Such a paradigm is brittle: (1) procedural code strictly depends on the order of operations, making generation prone to cumulative errors and cascading dependencies (Zhang et al., 2021; Xu et al., 2023a); (2) a linear representation diverges from human design intuition: engineers reason in terms of objects (parts/features) and their relations, rather than replaying every operation step by step (Stiny, 1980; Henderson, 2014); (3) procedural datasets often contain numerous complex shapes that are difficult to describe clearly in natural language (Wu et al., 2021; Willis et al., 2021), which reduces the interpretability and evaluability of text-to-CAD tasks; and (4) they encode many low-level geometric constraints (e.g., parallelism, concentricity, symmetry), which are useful in interactive software for maintaining geometric relations but largely redundant in code-based generation, where exact parameters can enforce them directly. What is truly valuable are high-level semantic constraints, such as "two holes must be concentric" or "gear teeth counts must match", which capture the logical dependencies between objects.

In contrast, an *object-oriented* modeling paradigm offers several advantages. It treats objects as the core units, each with attributes and methods, and supports the assembly of objects into complex structures. This hierarchical and modular representation more closely mirrors human design intuition (Stiny, 1980; Henderson, 2014), while naturally encoding semantic constraints between objects, thereby directly reflecting functional dependencies and design logic. In addition, object-oriented representations enable greater flexibility and robustness: modifying or replacing an object does not disrupt the overall structure, and maintaining an object library facilitates extensibility, reusability, and efficient error correction.

This shift is also consistent with recent studies that LLMs increasingly exhibit object-centric reasoning in complex tasks (Xu et al., 2023b; Wang et al., 2024; Kim et al., 2023). LLMs are not only capable of natural language generation, but can also organize knowledge and decompose tasks in an object-oriented manner. This type of reasoning closely aligns with human cognitive strategies for perception and reasoning (Marr, 1982; Carey, 2009), enabling more stable planning, composition, and abstraction at the object level. Therefore, continuing to rely on procedural CAD generation creates a misalignment with these natural reasoning tendencies, whereas adopting an object-oriented paradigm provides a better fit and unlocks the potential of LLM-driven design.

Motivated by these observations, we propose **OBJ2CAD**, a novel **object-oriented text-to-CAD generation framework**. OBJ2CAD represents CAD models as collections of objects, assembled into complex structures, while de-emphasizing redundant low-level constraints and highlighting logical relations between objects. To support this paradigm, we construct *a new dataset* designed around object-oriented representations and natural language describability. Building on this resource, we design *an LLM-driven framework* that combines top-down planning with bottom-up generation. In addition, we introduce *geometric assembly reasoning*, which explicitly transforms assembly tasks into geometric-mathematical constraint problems to enhance stability in spatial reasoning. Finally, we develop an *interactive iterative mechanism* that incorporates user feedback to refine local objects and incrementally expand the object graph, enabling continuous system evolution.

Our main contributions are as follows:

- The first framework OBJ2CAD, which is illustrated in Figure 1, to shift text-to-CAD from a procedural paradigm to an object-oriented paradigm.

- A new object-oriented dataset designed for natural language describability and hierarchical structure, de-emphasizing redundant low-level constraints while retaining essential semantic relations.

- An LLM-driven framework that combines top-down planning (task decomposition into objects and assemblies) with bottom-up generation (object instantiation and composition), providing a divide-and-conquer approach to text-to-CAD.

- Geometric assembly reasoning that formulates assembly as explicit geometric-mathematical constraints, improving spatial reliability.

- An interactive iterative mechanism that incorporates user feedback to expand the object graph, enabling continuous system improvement.

## 2 RELATED WORK

**Text-driven 3D generation.** Recent years have witnessed remarkable progress in text-to-3D generation. Early works such as CLIP-Mesh leverage CLIP guidance for zero-shot 3D generation (Khalid et al., 2022), while DreamFusion introduces Score Distillation Sampling (SDS) to distill pretrained diffusion models into 3D, achieving significant improvements in quality (Poole et al., 2022). Follow-up methods further advance resolution, representation richness, and stability, including Magic3D (Lin et al., 2023), Fantasia3D (Chen et al., 2023), and ProlificDreamer (Wang et al., 2023). DreamBooth3D (Raj et al., 2023) and MVDream (Shi et al., 2023) enhance personalization and multi-view consistency, while DreamGaussian proposes a Gaussian-splatting representation for efficient generation (Tang et al., 2023). More recently, Instant3D explores both sparse-view reconstruction and instant forward generation (Li et al., 2023; 2024), and MeshGPT employs autoregressive transformers to produce explicit meshes with sharper geometry (Siddiqui et al., 2024). Despite these advances, most approaches rely on *implicit representations* (e.g., NeRF, Gaussian splatting), which struggle to retain sharp edges and parametric precision crucial for engineering and CAD applications.

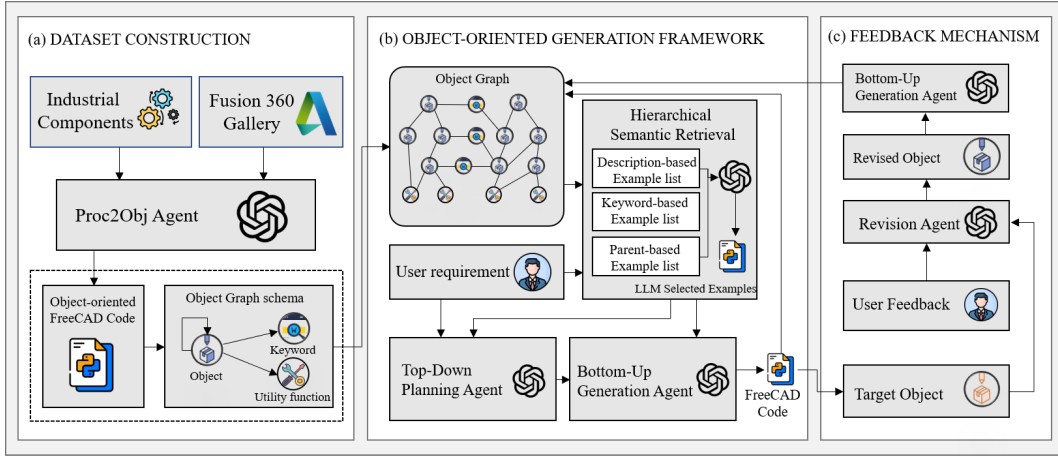

Figure 1: Overall framework of our proposed object-oriented text-to-CAD system. (a) **Dataset Construction:** We build a new dataset by converting industrial components and Fusion 360 Gallery models into object-oriented CAD code through the Proc2Obj Agent. The resulting dataset is organized into an object graph that links objects, keywords, and utility functions. (b) **Object-Oriented Generation Framework:** Given a user requirement, the system performs *Hierarchical Semantic Retrieval* to select relevant examples from the object graph (description-, keyword-, and parent-based retrieval). These examples are used to guide a two-stage generation process: a Top-Down Planning Agent for high-level decomposition and a Bottom-Up Generation Agent for executable FreeCAD code synthesis. (c) **Feedback Mechanism:** When errors occur, users can provide feedback targeting specific objects. A Revision Agent incorporates this feedback to produce a revised object, which is then re-integrated into the generation pipeline. The updated object is also stored in the object graph, enabling long-term improvement through accumulation of revised knowledge.

**Language-model-based CAD generation.** In parallel, language-driven CAD modeling has become an emerging direction. DeepCAD (Wu et al., 2021) and the Fusion 360 Gallery dataset (Willis et al., 2021) provide procedural construction sequences, enabling early attempts at learning CAD as a sequence prediction problem. SkexGen further models sketches and extrusions autoregressively with disentangled codebooks (Xu et al., 2022). More recently, Text2CAD directly generates CAD construction sequences from textual descriptions (Khan et al., 2024), while CADCodeVerify incorporates visual feedback to refine parametric CAD code using vision-language models (Alrashedy et al., 2025). Meanwhile, 3D-GPT demonstrates procedural generation of 3D content by prompting LLMs to synthesize Blender Python code (Sun et al., 2023). Although these methods highlight the potential of LLMs for CAD automation, they largely adopt a *procedural paradigm* that depends on long operation sequences, prone to error accumulation and redundancy in low-level constraints.

**Our approach.** In contrast, we propose an *object-oriented paradigm* for text-to-CAD, where CAD models are represented as hierarchical objects with semantic constraints. This shift reduces redundant operations, emphasizes design logic, and better aligns with both human design intuition and the object-centric reasoning capabilities of large language models.

## 3 METHODOLOGY

### 3.1 DATASET CONSTRUCTION

The most widely used benchmarks in text-to-CAD are DeepCAD (Wu et al., 2021) and the Fusion 360 Gallery Dataset (Willis et al., 2021), both derived from modeling logs of commercial CAD software. DeepCAD records sequences of sketches, extrusions, and Boolean operations from Onshape designs, while Fusion 360 preserves the complete design timeline of Autodesk Fusion 360 models. These datasets have been central to early CAD automation, enabling methods that map operation sequences to geometric outcomes. However, these datasets embody a procedural paradigm, where geometry is reconstructed step by step. For example, a part may begin with a 2D sketch, followed by extrusion, chamfering, mirroring, and Boolean combinations, with the final result strictly dependent on the operation order. This representation emphasizes *how* to construct rather than *what* the object is or *why* operations are applied, leading to several limitations:

- *Long-sequence dependency*: Even moderately complex parts may require hundreds of operations, where small errors compound and quickly derail generation.

- *Semantic gap*: Procedural traces describe isolated actions but fail to capture higher-level semantics such as object boundaries or assembly intent. Without this abstraction, natural language descriptions often cannot be aligned clearly to the underlying geometry, creating ambiguity in both interpretation and evaluation.

- *Redundant low-level constraints*: Datasets often encode large numbers of geometric constraints (e.g., parallelism, symmetry). While useful in interactive CAD, these constraints are unnecessary, or even distracting, in programmatic generation, where parameters can enforce them directly. What truly matters are semantic constraints that encode design logic.

- *Unfairness to LLMs*: Human designers rely on visual feedback to validate constraints, but LLMs trained only on operation logs must imitate sequences without grounding. This makes the task unnaturally difficult, limiting their potential for semantic alignment and object-oriented reasoning.

Therefore, while DeepCAD and Fusion 360 have been invaluable for early progress, their procedural representation is fundamentally misaligned with the reasoning capabilities of modern LLMs. To fully exploit object-centric reasoning and planning, a shift toward an object-oriented dataset is needed, which can organize designs hierarchically, minimize redundant low-level operations and constraints, and strengthen the alignment between natural language and geometric semantics.

**Our Dataset Design.** To address these limitations, we construct a new object-oriented text-to-CAD dataset, containing 1,000 samples. Unlike existing procedural datasets, ours centers on objects and their hierarchical structure, emphasizing linguistic describability and reusability. To make this representation precise, we first define how each object is formally described.

**Definition 1** (Object-Oriented CAD Representation). *Each object $o \in \mathcal{O}$ is defined as a function:*

$$o = (name, \theta, d, K, \text{body}) \tag{1}$$

*where* name *is the object name; $\theta$ is its parameter set; $d$ is a natural-language docstring describing its geometry and intended use; $K$ is a set of keywords for semantic retrieval; and* body *is the function body, with two forms:*

$$\text{body} = \begin{cases} \text{compose}(u_1(\theta), u_2(\theta), \dots), & \textit{Simple Object} \\ \text{assemble}(o_1, o_2, \dots, \mathcal{C}), & \textit{Composite Object} \end{cases} \tag{2}$$

*where simple objects are built directly from utility functions $u \in \mathcal{U}$, and composite objects are assembled from sub-objects with semantic constraints $\mathcal{C}$.*

This definition highlights the dual nature of our dataset: objects can either be simple units built from low-level utility functions, or composite structures recursively combining sub-objects under semantic constraints. To further capture the relationships among objects, keywords, and functions, we organize the dataset into a structured graph.

**Definition 2** (Object-Oriented CAD Graph). *The dataset is organized as a directed graph $G = (V, E)$, where*

$$V = \mathcal{O} \cup \mathcal{K} \cup \mathcal{U}, \qquad E = E_{contain} \cup E_{describe}. \tag{3}$$

*Here $\mathcal{O}$ are object nodes, $\mathcal{K}$ are keyword nodes, and $\mathcal{U}$ are utility function nodes. $E_{contain}$ denotes containment edges (object $\rightarrow$ sub-object or utility function), and $E_{describe}$ denotes description edges (object $\rightarrow$ keyword).*

Through this formalization, our dataset goes beyond raw object lists by explicitly encoding both compositionality and semantic alignment. To construct it, we develop the Proc2Obj agent, an LLM-driven system that converts procedural Fusion 360 CAD code into *object-oriented FreeCAD Python code*. Specifically, the agent first curates 900 shapes from the Fusion 360 Gallery, filtering for designs that are linguistically describable and excluding overly complex geometries that cannot be reliably expressed in natural language. It then translates these procedural code sequences into hierarchical object definitions in FreeCAD, preserving semantic constraints while removing redundant low-level operations. To complement this, we add 100 common industrial components (e.g., gears, bearings), authored with the assistance of LLMs, which naturally conform to the object-oriented paradigm. This combination ensures coverage of both everyday engineering parts and semantically clear exemplars with well-defined object boundaries. More technical details refer to Appendix A.

The resulting dataset offers three key advantages: (1) *Natural language alignment*: every object is paired with docstrings and keywords, bridging geometry and language; (2) *Hierarchical representation*: complex parts are composed recursively from simple ones, reducing long-sequence dependencies; (3) *Semantic focus*: redundant low-level constraints are de-emphasized, while high-level semantic constraints that capture design logic are retained, aligning better with LLM reasoning.

In this regard, the Proc2Obj agent provides a systematic way to transform procedural CAD code into object-oriented representations, creating a solid foundation for retrieval, planning, assembly reasoning, and interactive design. More than a dataset construction pipeline, it represents an essential step toward object-oriented semantic modeling for text-to-CAD.

## 3.2 OBJECT-ORIENTED GENERATION FRAMEWORK

The object-oriented dataset introduced in the previous section provides the representational foundation for text-to-CAD modeling, but data alone is not sufficient. To fully exploit the hierarchical structure and semantic information it encodes, we design a *object-oriented generation framework*, which combines example retrieval with LLM-driven planning and generation, forming a system that both aligns with human design thinking and maximizes the strengths of large language models.

The framework is motivated by three observations:

- LLMs excel at handling language and semantics but remain limited in spatial reasoning and geometric constraint solving. Procedural sequence generation amplifies these weaknesses through long-sequence error accumulation and redundant low-level constraints. In contrast, an object-oriented representation enables decomposition into object selection, planning, and assembly, allowing LLMs to focus on design logic rather than low-level operations.

- Human designers rarely create models entirely from scratch; they adapt prior designs. Similarly, LLMs benefit from example-based generation, which aligns with their few-shot nature and improves robustness.

- Separating planning from generation combines global design consistency with local accuracy. Top-down planning establishes the object hierarchy, while bottom-up generation incrementally instantiates and assembles components.

In practice, the framework operates as follows: given a user query, the system retrieves the most relevant example objects from the object graph, uses them to guide top-down planning of the object hierarchy, and then performs bottom-up generation, instantiating objects and solving geometric–mathematical constraints to complete the assembly. The following sections (Section 3.2.1 and Section 3.2.2) describe these two core components in detail.

### 3.2.1 TOP-DOWN PLANNING

In the object-oriented framework, the first step of generation is *planning*, which refers to inferring the set of objects and their hierarchical structure required to satisfy a user's natural language request. Unlike procedural generation that depends on replaying fixed sequences of operations, our planning process is closer to human design thinking: first identify the necessary parts and features, then determine how they should be combined and assembled.

To achieve this, we propose a *retrieval-guided planning* approach. The key idea is that while LLMs have limited ability in complex spatial reasoning, they are strong at analogy and example-based generalization. Thus, we transform planning into a process of *retrieval → reference → generation*, enabling the LLM to anchor its reasoning in semantically meaningful objects from the dataset rather than starting entirely from scratch. More specifically, a *hierarchical semantic retrieval* mechanism is formulated that integrates multiple semantic signals to improve robustness and relevance:

- *Description-based retrieval*: we compute the embedding of the user query $q$ and compare it with embeddings of object descriptions $d(o)$, selecting the top-$k$ most similar objects.

- *Keyword-based retrieval*: each object is associated with keyword nodes $k \in \mathcal{K}$. We retrieve objects connected to relevant keywords in the query and again select the top-$k$ by similarity.

- *Parent-based retrieval*: if the target object is part of a parent object, we additionally retrieve candidate sub-objects from the *children used in the parent's reference examples*, and select the top-$k$ most similar among them.

Formally, the overall candidate set is defined as:

$$\mathcal{R}(q) = \underbrace{\text{TopK}_{\mathcal{O}}\big(\text{sim}(f(q), f(d(o)))\big)}_{\text{Description-based}} \cup \underbrace{\text{TopK}_{\mathcal{K}}\big(\text{sim}(f(q), f(k))\big)}_{\text{Keyword-based}} \cup \underbrace{\text{TopK}_{\mathcal{P}(o)}\big(\text{sim}(f(q), f(d(o')))\big)}_{\text{Parent-based}} \quad (4)$$

where $f(\cdot)$ is the embedding function, $\text{sim}(\cdot, \cdot)$ is the similarity metric (e.g., cosine similarity), $\mathcal{O}$ is the set of object nodes, $\mathcal{K}$ is the set of keyword nodes, and $\mathcal{P}(o)$ is the set of sub-objects belonging to the parent of the target object. The three candidate pools are merged into a single retrieval set $\mathcal{R}(q)$, and the LLM selects a subset of examples from this set to guide planning. Through this mechanism, planning gains robustness, semantic grounding, and structural consistency.

It is worth noting that since our dataset is organized as a graph, it is naturally suitable for graph neural networks (GNNs). We experimented with GNN-based node representation learning to exploit structural context in retrieval. However, empirical results showed that GNNs did not yield significant improvements compared to simple embedding similarity, while incurring higher training and inference costs. For this reason, our final system adopts the simpler embedding-based retrieval method, striking a balance between effectiveness and efficiency.

### 3.2.2 BOTTOM-UP GENERATION

After the planning stage determines the object hierarchy, the system proceeds to bottom-up generation. Similar to top-down planning, this stage is also *retrieval-based*: we first retrieve example objects from the object graph that are most relevant to the target object or its sub-objects, and then use these examples as references to guide the LLM. The key difference is that generation here focuses not on *which* objects to include, but on *how* to accurately assemble them together.

To implement assembly, we adopt an explicit strategy of *semantic-to-mathematical constraint transformation*. In real-world scenarios, when user requirements or retrieved examples contain semantic descriptions such as "a support surface must be parallel to another", these are converted into geometric-mathematical forms like coincident centers, parallel normal vectors, or zero angle difference. These mathematical constraints are then provided to the LLM, which directly outputs geometric parameters (position, rotation, scale) that satisfy the constraints during generation.

The rationale behind this design is that while LLMs are still limited in direct spatial reasoning, they have been extensively optimized for mathematical reasoning tasks. For instance, prior work has shown that model developers employ techniques such as *chain-of-thought reasoning* and *process supervision* to significantly enhance performance on mathematical and logical benchmarks (OpenAI, 2023b; Liu et al., 2024). By explicitly converting assembly into a mathematical problem, we leverage these strengths of LLMs, enabling more accurate and stable assembly in CAD generation.

### 3.3 FEEDBACK MECHANISM

In CAD generation, especially for complex assemblies, it is unrealistic to expect a fully correct result in a single attempt. This difficulty arises not only from the inherent length and intricacy of CAD code, but also from the fact that design processes involve numerous fine-grained constraints and local variations. Applying feedback at the level of the entire code file is therefore both imprecise, since errors are hard to localize, and inefficient, as regenerating the full design is computationally costly and unstable. Our object-oriented framework naturally enables a more efficient and intuitive alternative. Instead of providing feedback on the whole design, users can intervene at the object level. When an issue is identified, the user specifies the problematic object node, and the system only needs to regenerate this object and propagate updates upward along the hierarchy. This localized revision drastically reduces correction cost and makes interactive design more practical.

Crucially, the mechanism also drives long-term improvement. Whenever a new or corrected object is created through interaction, it is automatically incorporated into the object graph:

$$o' = \text{RevisionAgent}(o, \text{feedback}), \qquad G' = G \cup \{o'\}. \quad (5)$$

As a result, future queries with similar requirements can directly reuse the updated object, avoiding repeated feedback. Over time, the accumulated revisions expand and refine the object library, providing richer references for retrieval and enhancing system stability and reusability.

Object-level feedback not only improves the *repairability* of single generations but also equips the system with the ability to "learn" from user interactions, evolving into a progressively more capable and robust assistant for CAD design.

## 4 EXPERIMENTS

### 4.1 DATASET CONSTRUCTION

To promote the transition of text-to-CAD from a procedural to an object-oriented paradigm, we construct a dataset of 1,000 object-oriented CAD examples. The generation is based on *FreeCAD Python scripts*: on the one hand, FreeCAD is open-source; on the other hand, compared to code-only frameworks such as CadQuery, FreeCAD additionally provides a full graphical user interface (GUI). Since CAD code generated by LLMs often requires further user modification and interaction, FreeCAD offers a more practical and user-friendly solution. The dataset design follows five principles: (i) geometry must be describable in concise and unambiguous natural language; (ii) shapes are organized hierarchically into objects and sub-objects rather than linear operation sequence; (iii) redundant low-level geometric constraints are removed, while high-level semantic constraints capturing design logic are retained; (iv) objects remain reusable and extensible as modular units; and (v) all examples are executable and verifiable through both automated and manual quality checks.

The dataset comprises two components. The first contains 100 common industrial parts such as gears, bearings, couplings, flanges, and keyed shafts. A small set of initial examples is manually authored and then expanded with GPT-5, with each result revised to ensure geometric correctness and semantic clarity. The second consists of 900 retrieval-driven examples from the Fusion 360 Gallery Reconstruction dataset. Candidates are chosen from rendered images, emphasizing designs that are linguistically describable and structurally decomposable, while filtering out overly complex or specialized geometries. For each candidate, the system retrieves similar objects from the first 100 parts as references, and GPT-5 generates new object definitions. Unsatisfactory outputs are corrected through human feedback or direct code modification, and all samples are manually validated for usability and consistency. Further details of the construction pipeline, including tool library maintenance, graph organization, and validation procedures, are provided in Appendix A.

### 4.2 COMPARISON OF LLM BACKBONES

To evaluate the effectiveness and generality of our framework OBJ2CAD, we conduct an experiment across different LLMs. We randomly sample 100 examples from our dataset as the evaluation set, while the remaining 900 examples are used to construct the object graph. For each test case, the

Table 1: Results of different LLM backbones under our proposed framework OBJ2CAD.

| Model | Compilation Success | Vision Consistency | Human Consistency |
|---|---|---|---|
| GPT-5 | **100%** | **85%** | **83%** |
| GPT-5-mini | 99% | 81% | 80% |
| LLaMA-3.2 90B | 85% | 70% | 66% |
| CodeLlama-70B | 83% | 64% | 60% |

LLM is guided by our framework to generate object-oriented CAD code. If the generated code fails to compile, the system returns the error message to the LLM and retries up to three times; only after three failures is the case considered unsuccessful.

We compare four representative LLMs: GPT-5, GPT-5-mini, LLaMA-3.2 Vision 90B, and CodeLlama-70B. Our objective is not to rank the raw capability of these models, but rather to demonstrate how well our framework can operate with different backbones. In this respect, we report results on three metrics: (1) *Compilation Success (%)* – percentage of test cases successfully compiled within three attempts; (2) *Vision Consistency (%)* – percentage of cases where the generated CAD model matches the ground-truth three views, automatically judged by GPT-5-mini with vision; and (3) *Human Consistency (%)* – percentage of cases where human evaluators judge the generated CAD model to match the ground-truth three views. The results in Table 1 show that OBJ2CAD performs consistently well with GPT-5 and GPT-5-mini, but degrades with open-source backbones like LLaMA-3.2 and CodeLlama, indicating that stronger language and reasoning abilities improve assembly reliability and vision alignment while the framework remains adaptable across models.

## 4.3 Controlled Comparison: OO vs. Procedural Paradigms

To directly evaluate whether the object-oriented (OO) representation provides measurable benefits over procedural generation independent of retrieval or prompting heuristics, we conduct a controlled experiment on the same set of 100 natural-language CAD prompts. We compare four systems, including **Procedural-Direct**: the LLM outputs a full CadQuery script in a single pass, producing a linear operation chain; **Procedural-Decomposed**: the LLM first produces a high-level task plan and then expands each step into CadQuery operations; **OO-ZeroShot**: the LLM generates FreeCAD's object-oriented Python structures without using any retrieval; and **Full Framework**: our complete object-oriented framework with hierarchical retrieval, planning, and refinement.

CadQuery is used for procedural variants due to its purely operation-sequence modeling interface, while FreeCAD naturally supports object-oriented variants via its hierarchical object tree. In all baselines, the LLM receives identical prompts, and retrieval is disabled to isolate the effect of the representation itself. The results in Table 2 show that both OO-

Table 2: Controlled comparison of procedural vs. object-oriented (OO) CAD generation paradigms.

| Method | Compilation Success | Vision Consistency | Human Consistency |
|---|---|---|---|
| Procedural-Direct | 83% | 35% | 30% |
| Procedural-Decomposed | 79% | 48% | 45% |
| OO-ZeroShot | 93% | 53% | 50% |
| Full Framework (Ours) | **100%** | **81%** | **80%** |

ZeroShot and our full framework deliver substantially more reliable and coherent generation than procedural baselines. This controlled setting demonstrates that the gains arise not from retrieval or prompt engineering, but from the representational advantage of OO paradigm.

This advantage primarily stems from the structural stability of OO representations. Procedural pipelines rely on fragile low-level geometric references and strict operation ordering, often leading to broken constraints, Boolean failures, or cascading pipeline collapse from a single mistake. In contrast, FreeCAD's object-level hierarchy isolates geometry into independent semantic units with explicit attributes and parent-child relations, providing far more robust generation behavior. Furthermore, OO structures align better with human design semantics and naturally support retrieval, reuse, and compositional synthesis. These properties enable our full framework to achieve the highest vision and human consistency scores, demonstrating that OO paradigm produces markedly more reliable, semantically coherent, and reusable CAD programs than procedural approaches.

## 4.4 User Feedback Evaluation

To evaluate the effectiveness of our feedback mechanism, we construct a 25-example test set evenly divided into five groups: (1) single-object designs, (2) two-object designs, (3) medium designs with 3–5 objects, (4) complex designs with 6–10 objects, and (5) highly complex designs with more than 10 objects. This stratification captures increasing difficulty of CAD generation as semantic reasoning and assembly constraints grow with object count.

A generation is considered *successful* if the target design is completed within five feedback interactions. Two quantitative measures are reported: (1) *Successful Cases (SC)* – the number of test cases solved within the feedback budget, and (2) *Average Feedback (AF)* – the mean number of feedback rounds required among successful cases. This

Table 3: Performance of our full framework and ablation variants across object-count regimes. Each group has **SC** (Successful Cases) and **AF** (Avg. Feedback, rounds).

| Method | 1 obj | | 2 objs | | 3–5 objs | | 6–10 objs | | >10 objs | |
|---|---|---|---|---|---|---|---|---|---|---|
| | SC | AF | SC | AF | SC | AF | SC | AF | SC | AF |
| GNN-based Retrieval | 5 | 1.2 | 5 | 1.6 | 5 | 2.0 | 4 | 2.8 | 3 | 2.8 |
| w/o Math Conversion | 5 | 1.6 | 5 | 2.0 | 4 | 3.2 | 3 | 3.2 | 2 | 3.4 |
| Nearest Embedding Only | 5 | 1.8 | 5 | 2.2 | 3 | 3.6 | 2 | 3.4 | 2 | 4.0 |
| Full Framework (Ours) | 5 | 1.2 | 5 | 1.4 | 5 | 2.0 | 4 | 2.6 | 4 | 2.8 |

dual metric captures both generation reliability and user interaction efficiency. Results are summarized in Table 3. Across all complexity levels, the system achieves high success rates, while requiring only one or two targeted object-level modifications. These results highlight that our framework enables efficient, targeted repair without regenerating entire designs, significantly improving the practicality and usability of complex CAD generation.

## 4.5 Ablation Study

To further assess the contribution of each component in our framework, we conduct an ablation study on the same 25-example test set. We compare the full system against three simplified variants: (1) *w/o Math Conversion*: removes geometric-mathematical constraint formulation, relying only on semantic reasoning; (2) *GNN-based Retrieval*: replaces our embedding-based retrieval with a

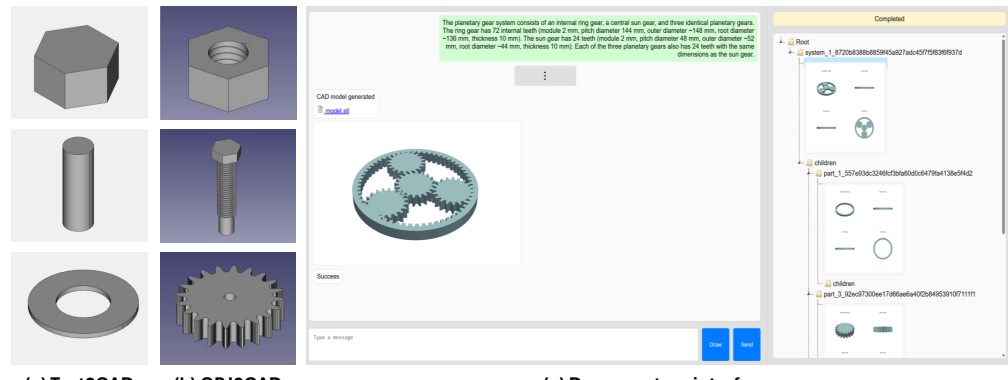

|  |  |  |
|---|---|---|
| (a) Text2CAD | (b) OBJ2CAD | (c) Demo system interface |

Figure 2: Case study by comparing our object-oriented Obj2CAD with procedural Text2CAD on three industrial parts (hexagonal nut, hex bolt, and involute gear), and the interface of our demo system: (a) Text2CAD outputs, (b) Obj2CAD outputs, and (c) demo system interface.

GNN encoder trained on the object graph; and (3) *Nearest Embedding Only*: selects top-$k$ nearest examples purely by embedding similarity without multi-source retrieval or LLM re-ranking.

Table 3 reports results under the evaluation protocol described in Section 4.4. The full framework consistently achieves the highest success rates while requiring fewer feedback rounds. Notably, removing the math conversion leads to a clear drop in assembly accuracy, confirming the importance of leveraging LLMs' enhanced mathematical reasoning ability. The GNN-based retrieval shows no significant advantage over simple embeddings, while incurring additional computational cost. Finally, the nearest-embedding baseline performs worst, indicating that our hierarchical semantic retrieval combined with LLM re-ranking is crucial for reliable and efficient planning and generation.

### 4.6 CASE STUDY AND DEMO DEMONSTRATION

We qualitatively compare our object-oriented OBJ2CAD against procedural Text2CAD (Khan et al., 2024) and several similar open-source CAD systems on our benchmark. However, these models are trained on small procedural operation-sequence corpora using small-scale open-source LLMs, and therefore only capable of producing very simple geometries. As illustrated in Figure 2(a), Text2CAD fails to generate valid or complete geometry even for fundamental mechanical parts. In most cases, the generated code does not compile or produces severely incomplete shapes, making it impossible to evaluate the method under our metrics. Further discussion is detailed in Appendix C.

To further illustrate the practicality of our framework, we have developed an interactive demo system (Figure 2). The demo[1] allows users to input textual requirements, retrieve examples, and generate object-oriented CAD models, which are displayed together with their hierarchical object graph and corresponding FreeCAD code. As shown in the planetary gear example, the system can generate not only the final CAD model but also the intermediate object structure and assembly relations, enabling step-by-step inspection and revision. Moreover, the generated design is exported in the native `.FCStd` format, allowing users to download and further edit the model directly in FreeCAD, thereby bridging automatic generation with human-in-the-loop refinement. For clarity, the screenshot in Figure 2 omits intermediate processes such as planning steps and feedback interactions. Appendix B provides more complex cases and additional details of the demo system.

### 5 CONCLUSION

This work introduces OBJ2CAD, an LLM-driven framework that advances Text-to-CAD from a procedural to an object-oriented paradigm. By constructing a new dataset of 1,000 object-oriented CAD examples and designing a generation pipeline that integrates retrieval, top-down planning, bottom-up generation, geometric assembly reasoning, and interactive feedback, OBJ2CAD aligns CAD synthesis with both human design intuition and the reasoning strengths of large language models. Experiments demonstrate its effectiveness across multiple LLM backbones and highlight the benefits of object-oriented representations for semantic alignment and assembly reliability. Future work will expand the dataset, integrate multi-modal inputs, and explore broader design applications.

---

[1]Demo and dataset available at `https://github.com/objcad5-hub/obj2cad`

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

## A  DATASET CONSTRUCTION DETAILS

In this section, we provide additional details of how the dataset was constructed, complementing the high-level description in the main text.

**Industrial Part Set.**   The construction process began with 100 common industrial parts, including gears, bearings, couplings, flanges, and keyed shafts. Initially, a small number of examples were written manually to establish templates. These served as reference points for subsequent expansions with GPT-5. During generation, the researcher actively intervened by revising the code whenever the output deviated from expected functionality or geometry. Once a part was successfully verified, it was added into the pool of reference components. Over time, this iterative procedure enabled a gradual bootstrapping of the dataset, where earlier verified examples improved the quality of later ones through retrieval-based prompting.

**Retrieval-Driven Expansion.**   The remaining 900 examples were sourced from the Fusion 360 Gallery Reconstruction dataset. Candidate designs were selected based on rendered images rather than raw procedural logs, prioritizing those that could be naturally described in language and structurally decomposed into sub-objects. Overly complex or highly specialized designs were filtered out. For each candidate, the system retrieved the most similar examples from the 100 industrial parts to serve as demonstrations in the prompt. GPT-5 then generated a new object definition, which was further refined through either direct human feedback or manual code edits. Only when both semantic clarity and geometric validity were confirmed was the example added into the dataset.

**Utility Function Library.**   A central component of the dataset construction is the dynamic maintenance of a utility function library. These functions, such as `wire_from_points`, `face_from_wire`, or `extrude_profile`, act as the atomic building blocks of object construction. Unlike traditional datasets with a fixed set of primitives, our approach maintains an evolving library that grows over time:

- Before generating a new object, GPT-5 selects 20 candidate functions from the library and incorporates them into the prompt. This balances efficiency (by keeping the prompt length manageable) and relevance (by focusing on the most likely functions).

- If the model requires a function not included in the current selection, it generates a new one. The candidate function is then compared against the existing library, and only added if it is not redundant.

- This process allows the library to expand dynamically while remaining compact, ensuring that new functions are introduced only when necessary and that duplicates are avoided.

Through this iterative process, the library evolves in tandem with the dataset, forming a consistent and extensible foundation for object-oriented CAD modeling.

**Graph Representation.**   The dataset is organized as a heterogeneous graph to explicitly encode relationships between different elements. The graph contains three types of nodes: object nodes, keyword nodes, and utility function nodes. Edges are defined as follows:

- *contain:* representing parent–child relationships among objects, capturing hierarchical composition;

- *described_by:* linking objects to keyword nodes that describe them linguistically;

- *uses:* connecting objects with the utility functions required for their construction.

Within this structure, complex objects assemble sub-objects through high-level semantic constraints, such as "hole–shaft concentricity" or "face alignment." These constraints are preserved both at the abstract level (as part of the object graph) and at the executable level (as concrete geometric–mathematical conditions).

**Validation and Quality Assurance.**   To ensure dataset reliability, we employed a two-stage validation procedure:

---

**Algorithm 1:** Construction of the Object-Oriented CAD Dataset

---

**Input:** Fusion 360 Gallery dataset $F$; initial industrial part set $I$
**Output:** Object-oriented CAD dataset $\mathcal{D}$ and object graph $G$

**Class: Tool Library Maintenance** ;
**class** ToolLibrary: ;
   $\mathcal{U} \leftarrow \emptyset$ ;                       `// Initialize tool library`
   `select_tools`$(query, k)$`:` ;
      Compute similarity between $query$ and tools in $\mathcal{U}$; return top-$k$ tools ;
   `update_library`$(u)$`:`;
      If $u \notin \mathcal{U}$ (checked by semantic deduplication), then add $u$ into $\mathcal{U}$ ;

**Main Process** ;
Initialize $\mathcal{D} \leftarrow \emptyset$ and $\mathcal{U} \leftarrow$ ToolLibrary();
**foreach** *industrial part $i \in I$* **do**
   | Generate object definition with GPT-5 and manually revise ;
   | $\mathcal{D} \leftarrow \mathcal{D} \cup \{i\}$ ;
   | $\mathcal{U}$.update_library(new tools appearing in generation) ;

Filter 900 candidates $C$ from $F$ by linguistic describability ;
**foreach** *candidate $c \in C$* **do**
   | $T \leftarrow \mathcal{U}$.select_tools($c$, 20) ;
   | Retrieve similar objects from $\mathcal{D}$, combine with $T$ as prompt ;
   | Generate new object $o$ with GPT-5, revise with human feedback ;
   | $\mathcal{D} \leftarrow \mathcal{D} \cup \{o\}$ ;
   | $\mathcal{U}$.update_library(new tools) ;

Construct object graph $G = (V, E)$: nodes are objects/keywords/tools, edges are `contain`, `described_by`, `uses`;
**return** $\mathcal{D}, G, \mathcal{U}$ ;

---

- **Automated checks:** verifying deterministic reproducibility across multiple runs, validating that assembly constraints are satisfied, and ensuring that the utility function library remains free of duplicates.

- **Manual checks:** reviewing object names for intuitiveness, confirming that natural language descriptions are unambiguous, and validating that assembly semantics are consistent with engineering intuition.

When a generated object failed to meet these standards, it was revised iteratively, either by providing targeted feedback to GPT-5 or by direct manual editing, until it passed both automated and manual validation. This feedback-and-revision loop was essential to maintain the semantic and geometric quality of the dataset.

**Outcome.** By combining retrieval-driven generation, dynamic tool library maintenance, heterogeneous graph representation, and rigorous validation, we constructed a dataset that is both semantically aligned with human intuition and structurally tailored for object-oriented CAD modeling. The resulting 1,000 examples provide a strong foundation for evaluating retrieval, planning, generation, and feedback mechanisms in downstream tasks.

## B   DEMO DETAILS

Figures 3–7 present a complete walkthrough of our interactive demo system. Starting from natural language requirements, the system generates hierarchical plans, produces part-level shapes, and integrates iterative feedback for correction (e.g., adding missing holes). Each stage is visualized step by step, showing how plans are translated into geometry, how errors are identified and fixed, and how the final assembly is produced. These examples highlight the transparency and interactivity of the proposed framework, as well as its ability to refine designs incrementally without restarting the entire process.

> I want design a Storage rack 8 feet wide, 6 feet long and 5 feet high, there should be 6 storage areas within it, and the rack should have a caster wheel in each corner.

**Plan_generated**

Overall Assembly Plan

1. Caster
Sub-Components:

Mounting Base:
Composed of a main base plate and a steer adapter. It provides a connection interface for attaching the caster to the frame.

Swivel Assembly:
Composed of a fork bracket (the U-shaped support structure) and a swivel ring. This assembly enables the caster wheel to rotate in any direction.

Rolling Wheel:
Consists of a hub and a tire. The hub supports the tire, and the tire provides the rolling function.

Assembly Approach:

The individual parts of a caster wheel are modeled separately with functions (such as makeMountingBase(), makeSwivelRing(), makeForkBracket(), and makeWheel()).

A complete caster module is obtained by fusing these parts using a function (for example, makeCasterWheel()).

Four caster modules are generated and then positioned at the four corners under the frame.

2. Frame Component
Sub-Components:

Columns (Vertical Posts):
Four posts located at the four corners of the frame. These provide vertical support and overall rigidity.

Perimeter Beams (Frame Members):
Eight beams in total, with four beams at the bottom and four beams at the top. These beams connect the posts and form the outer rectangular frame.

Intermediate Shelf:
A horizontal board that divides the storage rack into two main layers. It also adds stiffness to the overall structure.

Additional Horizontal Shelves (Lower and Upper):
One lower shelf is installed near the bottom and one upper shelf is installed within the top compartment. These help further segregate the internal space.

Internal Vertical Dividers:
Two dividers per layer (four in total) placed along the width to divide each layer into three storage compartments.

Back Panel (Optional):
A back panel may be added to improve torsional stiffness and enhance aesthetics. It is mounted on the rear side of the rack.

Assembly Approach:

Each sub-component is modeled in its own function with local parameters (dimensions, placements, etc.).

The overall frame is obtained by using Boolean fusion (fuse) of the sub-components in a main function (for example, makeRackFrame()).

> This plan is OK, we will continue

**Plan_generated**

1. Mounting Base:

Used to fix the entire universal wheel, with mounting holes for easy connection with the frame.

The Python script can be modeled as a flat plate (or a three-dimensional base structure) with holes reserved.

2. Swivel Assembly:

Realize the omnidirectional rotation function of the universal wheel.

This part mainly constructs a steering bearing (or shaft), which you can consider as a hollow cylinder or a geometric body with a sliding/rotating interface.

A simple chimeric structure can usually be added to ensure that the external rolling wheel body can rotate along this steering mechanism.

3. Rolling Wheel:

The wheel body can be further divided into two sub-parts:

a. Wheel Hub: The internal load-bearing structure provides the connection between the wheel body and the steering mechanism.

b. Tire: Surrounding the outer layer of the wheel hub, providing contact and friction with the ground. If necessary, it can be directly designed as a ring or tubular entity with outer patterns.

In the Python script, it is recommended to write two functions to generate the hub and tire respectively, and then use Boolean operations (such as "fusion") to merge them into a complete wheel model.

> Plan is OK, continue

Figure 3: Demo example (1): system interface showing natural language input, generated plan, and corresponding CAD output.

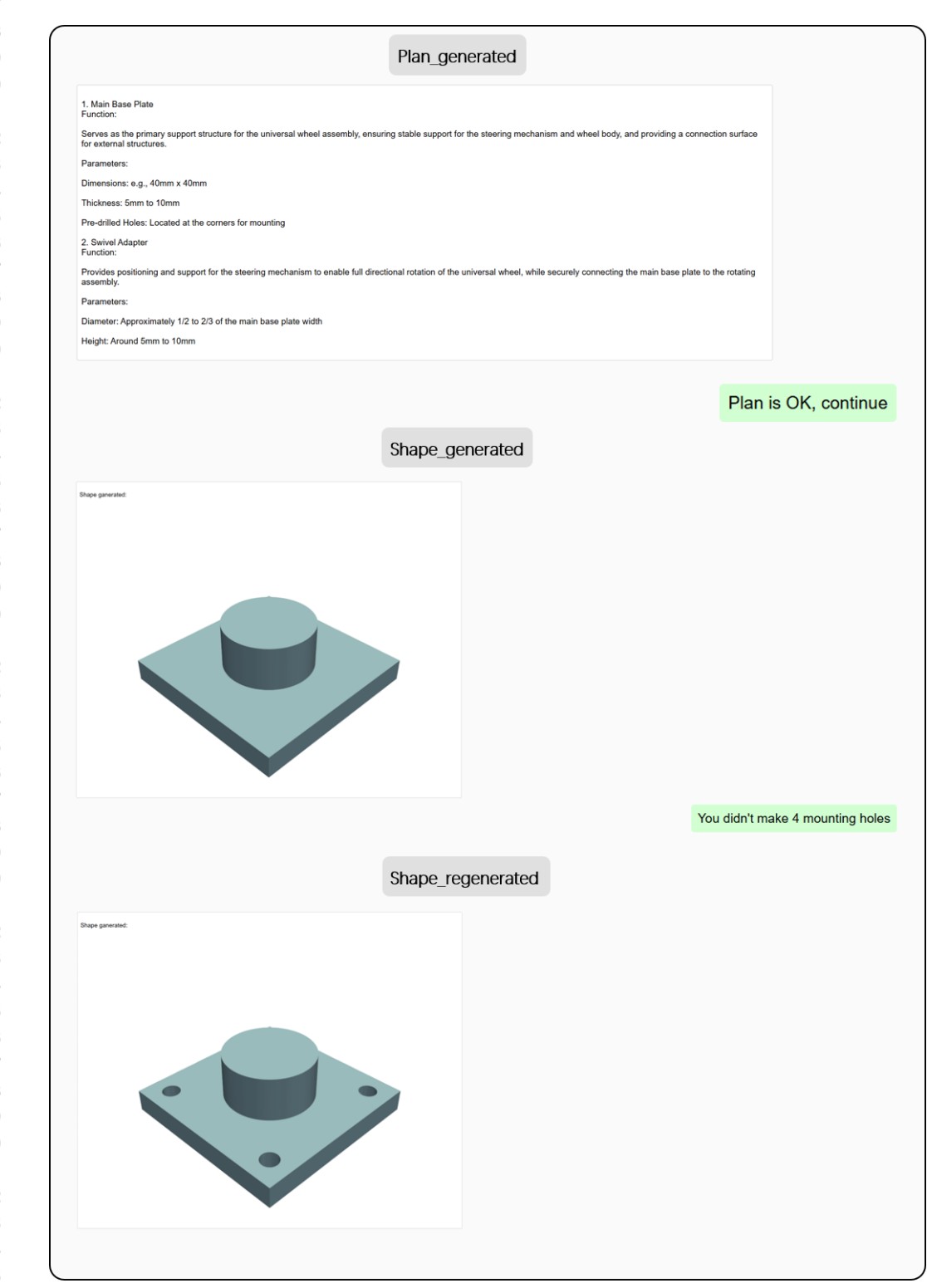

Figure 4: Demo example (2): detailed component generation with user feedback applied to correct design issues.

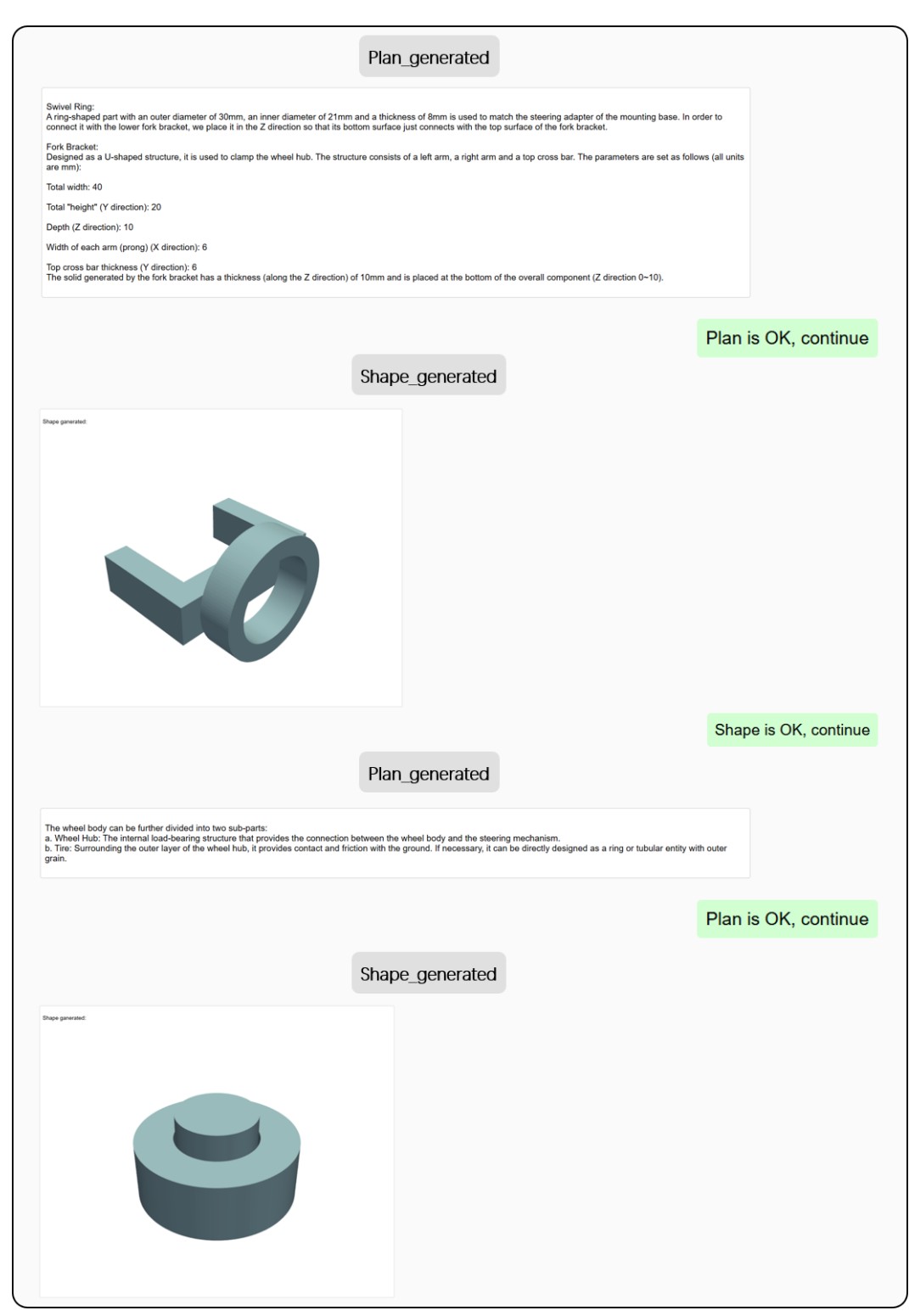

Figure 5: Demo example (3): iterative refinement of parts and integration into the overall object graph.

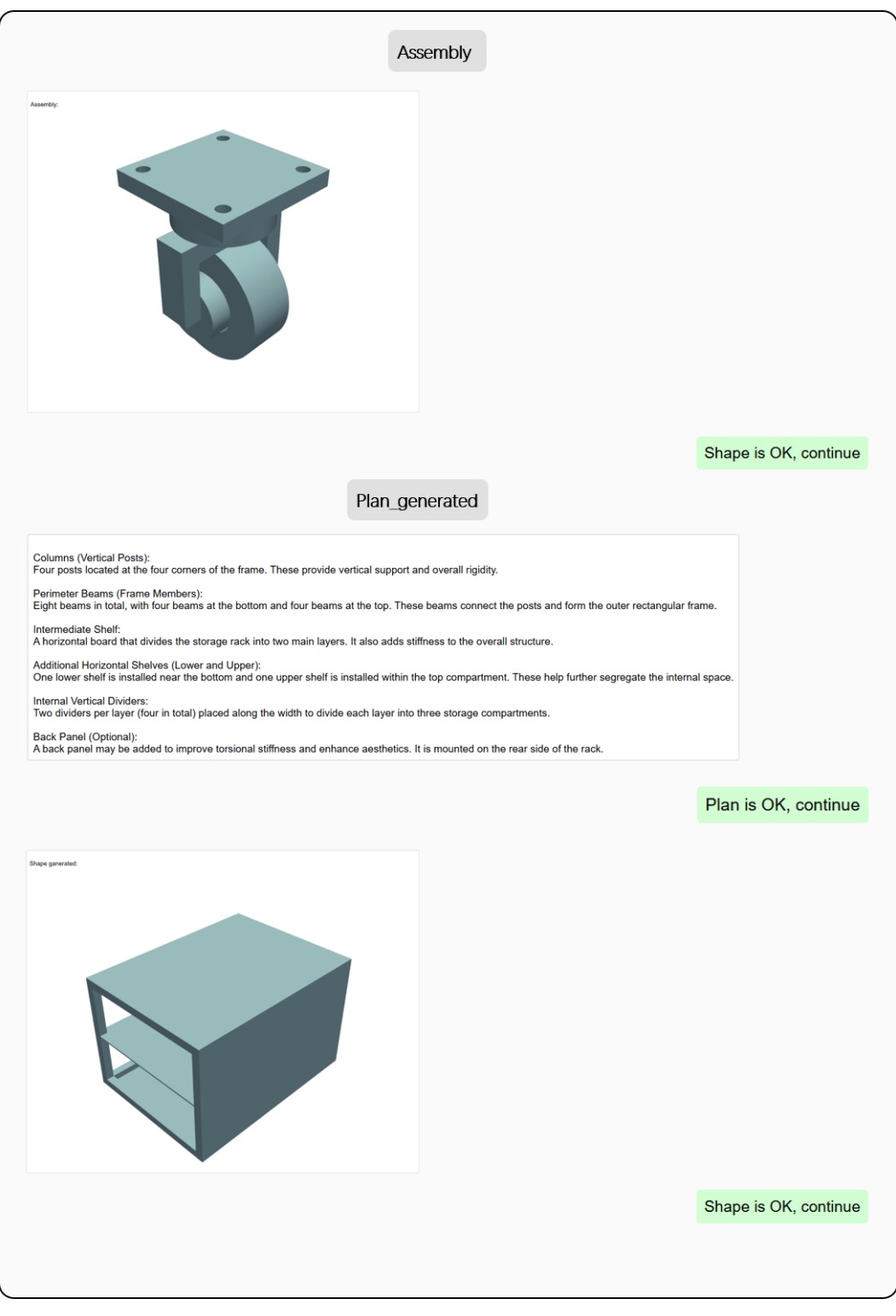

Figure 6: Demo example (4): progressive assembly of generated components into a coherent CAD structure.

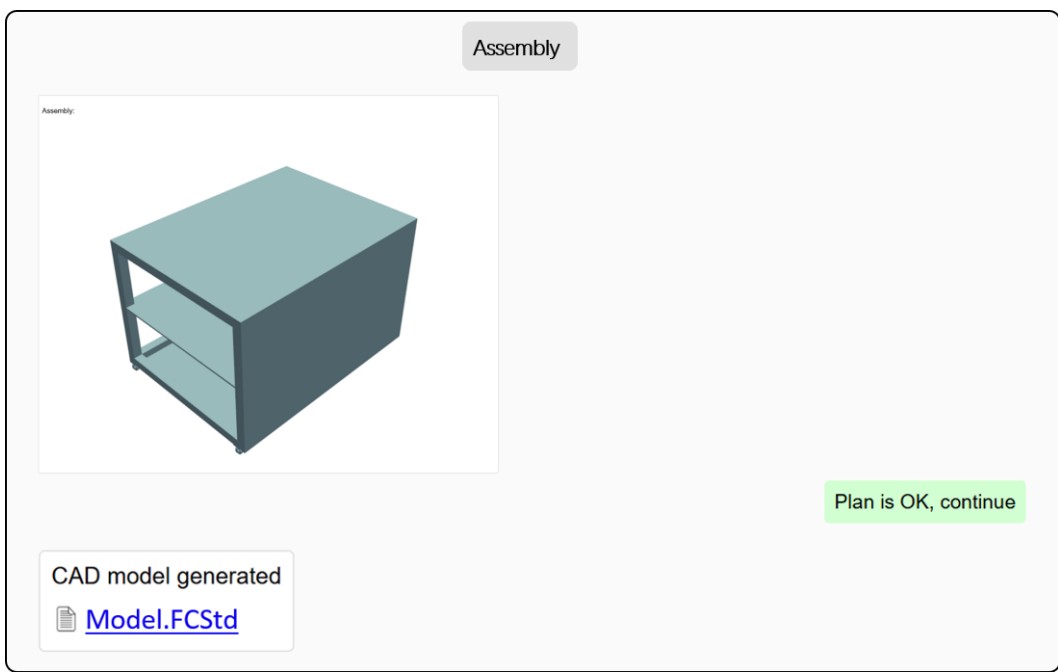

Figure 7: Demo example (5): final assembly result of the storage rack, with CAD model downloadable in `.FCStd` format for FreeCAD editing.

It is worth noting that this demo includes a representative failure example. The system first produces a complete and semantically correct high-level plan including "four mounting holes". However, in the first round of shape generation, the LLM instantiates only the base plate and the central adapter, omitting all four mounting holes. This type of failure occurs when the LLM under-expands object definitions or fails to instantiate some sub-objects. In our framework, such errors are typically corrected through iterative feedback and localized regeneration as shown in the demo, where the second iteration correctly adds the holes.

For more complex parts, OBJ2CAD failures more frequently stem from incorrect spatial relationships between sub-objects. Typical issues include: inaccurate inference of relative positions, misalignment or unintended gaps, incorrect parameters for arrays, rotational features, or tooth profiles, and mistakes in enforcing concentric, symmetry, or alignment constraints. Although our math conversion mechanism improves geometric reasoning, multi-part assemblies with strong spatial constraints can still exceed the LLM's reasoning capacity.

## C  COMPARISON WITH PROCEDURAL TEXT2CAD

In Section 4.6, we qualitatively compare our object-oriented framework OBJ2CAD against Text2CAD (Khan et al., 2024) and several similar open-source procedural CAD systems on our benchmark. Ad discussed, these procedural models are trained on small operation-sequence corpora using lightweight open-source LLMs, which limits them to producing only very simple geometries. As illustrated in Figure 8, Text2CAD fails to generate valid or complete geometry even for fundamental mechanical parts such as a hexagonal nut with an internal thread, a hex bolt with a hex head and a threaded shaft, and an involute spur gear. In most cases, the generated code either does not compile or produces severely incomplete shapes, making it impossible to evaluate the method under our metrics (compilation success, vision consistency, human consistency). Because Text2CAD cannot produce a sufficient number of valid outputs to support consistent measurement, meaningful quantitative comparison is not feasible.

It is important to note that even zero-shot code generation using a modern commercial LLM already exceeds the capabilities of Text2CAD by a substantial margin, and our object-oriented framework

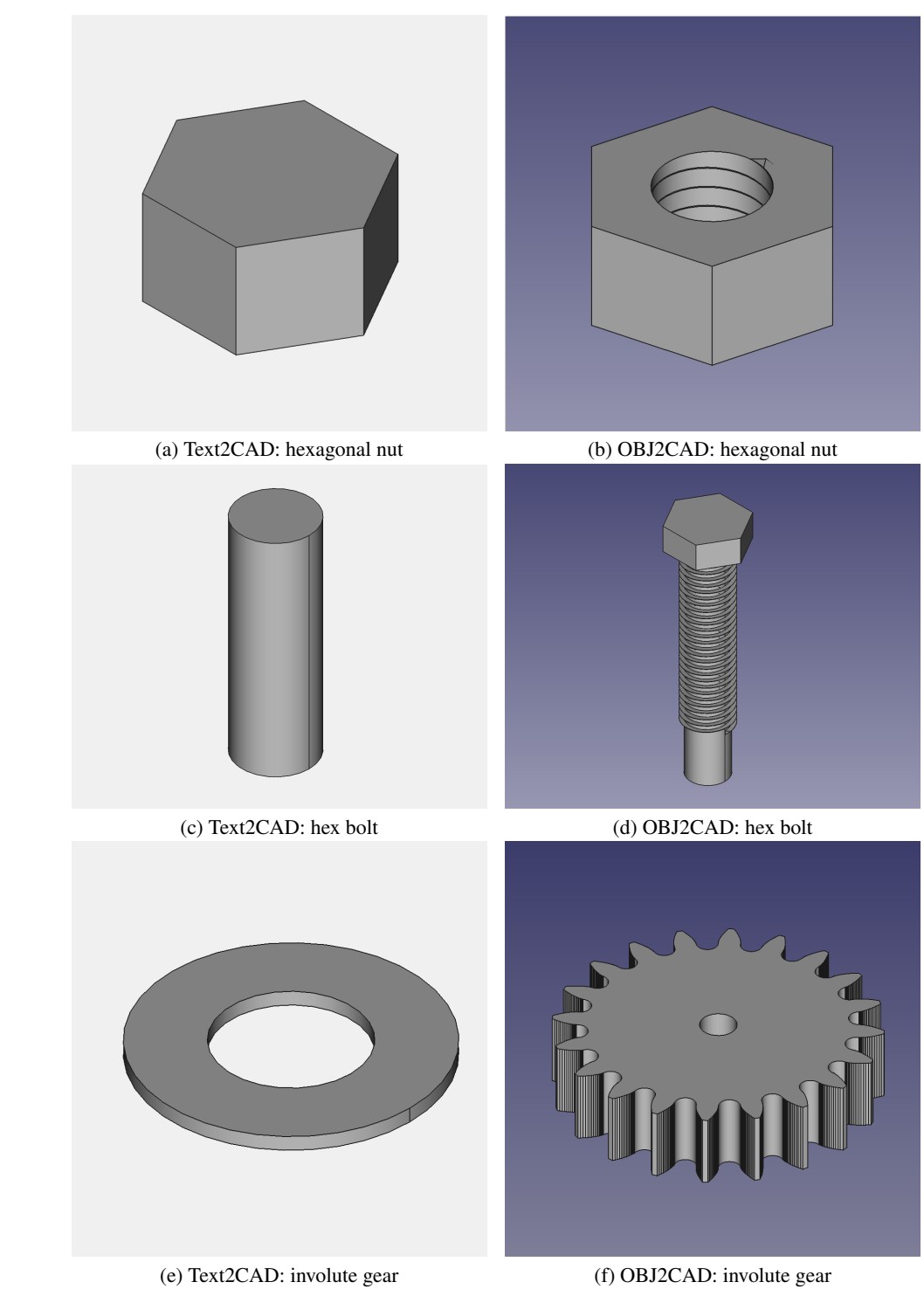

(a) Text2CAD: hexagonal nut          (b) OBJ2CAD: hexagonal nut

(c) Text2CAD: hex bolt          (d) OBJ2CAD: hex bolt

(e) Text2CAD: involute gear          (f) OBJ2CAD: involute gear

Figure 8: Comparison between Text2CAD (left column) and OBJ2CAD (right column) on three common industrial parts: hexagonal nut, hex bolt, and involute gear.

further improves reliability and structural coherence beyond that. For these reasons, we exclude Text2CAD from the main quantitative evaluation, as its capability ceiling is not aligned with the complexity of our benchmark or with the minimum requirements of industrial CAD modeling.

## D    COST ANALYSIS

We conduct a detailed cost analysis over the 100 samples used in Section 4.2, covering both full-generation and feedback stages.

- **Cost of generating a complete CAD part**. Across the 100 samples, the average token usage per sample is 35,156.12 input tokens and 32,854.64 output tokens. Using GPT-5-mini, this corresponds to approximately $0.074 per sample, and even with GPT-5, the cost remains modest at $0.37 per sample. Thus, the end-to-end generation is inexpensive and scales well.

- **Cost of the feedback mechanism**. During feedback iterations, we only submit the high-level plan, the relevant object code, and the feedback message to the model. Across our experiments, each feedback round uses roughly 10K input tokens and 5K output tokens on average. This results in a cost on the order of a few cents per iteration with GPT-5-mini, and still well below the cost of a full regeneration pass even when using GPT-5.

Since both the full-generation and iterative feedback stages operate with moderate token budgets, the use of closed-source LLMs in OBJ2CAD is cost-effective and operationally feasible.

## E    USE OF LARGE LANGUAGE MODELS

During the preparation of this paper, we made use of large language models (LLMs), such as GPT-5, primarily to assist with language editing and refinement. Specifically, LLMs were used to improve grammar, polish writing style, and rephrase sentences for clarity and readability. All core contributions of the work — including research ideas, dataset construction, method design, experimental setup, and result analysis — were independently conceived and implemented by the authors. The role of LLMs was limited to improving the presentation of the text, ensuring the paper meets academic writing standards.

