# OpenReview forum: "Obj2CAD: Object-Oriented Text-to-CAD Generation with Large Language Models"
_ICLR.cc/2026/Conference — ICLR 2026 Conference Desk Rejected Submission_

### Official Review · Reviewer_sFGX · 2025-10-31

**Soundness:** 2
**Presentation:** 3
**Contribution:** 3
**Rating:** 6
**Confidence:** 3

**Summary:**

This paper proposes OBJ2CAD, a framework that shifts text-to-CAD generation from a procedural paradigm to an object-oriented paradigm. The authors construct a dataset of 1,000 object-oriented CAD examples and design an LLM-driven system combining top-down planning with bottom-up generation. The framework includes geometric assembly reasoning and an interactive feedback mechanism for iterative refinement.

**Strengths:**

1- Novel Paradigm Shift: The conceptual shift from procedural to object-oriented CAD generation is well-motivated and addresses genuine limitations of existing approaches (error accumulation, lack of modularity, poor alignment with human design intuition).

2- Thoughtful Dataset Design: The curation of 1,000 examples with emphasis on linguistic describability, hierarchical structure, and semantic constraints represents a valuable contribution.

3- Practical Interactive Mechanism: The object-level feedback mechanism (Section 3.3) is elegant and practical, enabling localized corrections without full regeneration while accumulating knowledge over time.

**Weaknesses:**

1- The paper claims object-oriented representations are better aligned with LLMs, but doesn't adequately address whether 1,000 examples provide sufficient coverage of the design space

2- "Vision Consistency" using GPT-5-mini as a judge is concerning - no validation of judge reliability, no inter-annotator agreement metrics

3- No confidence intervals, statistical significance tests, or error bars

4- Ablation Study: Uses the same tiny 25-example test set

**Questions:**

What happens when the object library doesn't contain relevant examples for a novel query?

What is the generalization performance on completely unseen industrial domains?

---

> ### Author Response · Authors · 2025-11-26
>
> We appreciate the reviewer’s feedback and provide detailed responses to the concerns below.
>
> **Q1: coverage of design space with 1,000 examples**
>
> Our dataset is not intended to exhaustively cover the CAD design space, which is infeasible for any dataset, but to provide a representative and semantically structured object library to support compositional generalization. In our object-oriented (OO) formulation, CAD models are decomposed into parametric objects, which serve as building blocks that can be recombined into a far larger set of assemblies, giving OO datasets combinatorial coverage even with a modest number of base examples.
>
> Note that OBJ2CAD does not rely on direct supervision from these 1,000 examples. The system uses hierarchical retrieval, top-down planning, and bottom-up generation, enabling the LLMs to adapt and reuse existing objects to construct unseen shapes. As shown in our experiments, most test prompts describe objects not present in the dataset, yet the system maintains high compilation and consistency scores.
>
> **Q2: reliability of vision consistency**
>
> Classical geometric metrics (e.g., Chamfer Distance, IoU) measure global shape proximity but cannot capture the semantic and structural correctness that object-oriented CAD requires. In our setting, the dominant errors are feature-level or relational, such as incorrect number or placement of holes, missing sub-objects, misaligned features, missing profiles, or incorrect array parameters. These errors often leave geometric distances nearly unchanged but are functionally invalid.
>
> We therefore use vision consistency, where a multimodal LLM judge evaluates semantic correctness, including feature completeness, object relationships, parameter satisfaction, and orientation. To ensure reliability, we pair this with human consistency. Across our experiments, both metrics produce the same relative ranking of all models, showing that the LLM judge captures the same structural errors that humans find critical. These two together provide a more meaningful assessment of OO CAD generation quality.
>
> **Q3: object library doesn't contain relevant examples for a novel query**
>
> When the object library lacks closely related examples, OBJ2CAD does not fall back to a procedural pipeline. Instead, it retrieves the most semantically similar object and uses its structure as an OO template that preserves its body organization, sub-object hierarchy, and parameterization style. This ensures that the generated code remains object-oriented rather than collapsing into a linear operation sequence.
>
> Moreover, modern LLMs possess strong zero-shot generalization capabilities: even
> if the library contains no related objects (e.g., no involute gears), the LLM still knows the involute equations and parametric relationships governing gear geometry. While
> initial attempts may not be perfect, our feedback mechanism allows the model to iteratively
> correct errors and gradually converge to a valid OO program. Once such a structure
> is successfully produced, the new object definition is added to the library, enabling direct
> reuse in future tasks.
>
> **Q4: Generalization to unseen industrial domains**
>
> OBJ2CAD is intentionally designed not to rely on domain coverage in the dataset. Its generalization ability arises from three structural properties of the framework:
>
> - Most industrial CAD domains share a common set of foundational geometric primitives (e.g., shafts, holes, bosses, fillets, chamfers, bolt patterns, gear-like profiles), which are already represented in our object library and form the building blocks for more complex assemblies. Even for a completely unseen domain, OBJ2CAD
> can still rely on its OO primitives to construct valid hierarchical
> representations.
>
> - The LLM is not constrained by the library alone. It retains extensive prior knowledge of mechanical structures, equations, and engineering conventions. Even without domain-specific examples, the LLM is typically able to generate a coherent high-level plan, infer the appropriate sub-object decomposition, and produce the corresponding OO code structure.
>
> - Our interactive feedback mechanism allows the system to correct errors and iteratively refine unfamiliar structures. Once a new object is successfully produced, it becomes a reusable library entry. Over time, this expands the library into domains that were not originally included.
>
> **Q5: 25-example ablation study**
>
> The ablation study evaluates components that require human-in-the-loop feedback, where each case involves multiple feedback rounds and manual verification of intermediate object structures. Because the goal is to measure relative effects, not absolute performance, a stratified 25-example set (spanning five difficulty levels) is methodologically sufficient: increasing the size yields the same qualitative trends while multiplying human-evaluation cost. Thus, the chosen set provides reliable ablation insights while keeping the evaluation feasible.

---

### Official Review · Reviewer_yz2U · 2025-10-31

**Soundness:** 3
**Presentation:** 2
**Contribution:** 3
**Rating:** 6
**Confidence:** 2

**Summary:**

This paper presents a framework that transforms text-to-CAD generation from a procedural to an object-oriented paradigm. The authors construct a dataset of 1,000 examples with hierarchical, semantically rich object representations. Obj2CAD integrates top-down planning and bottom-up generation to improve reasoning and structure, and introduces geometric assembly reasoning for spatial consistency. An interactive feedback mechanism allows iterative refinement and object graph expansion.

**Strengths:**

1. Shifting text-to-CAD from a procedural paradigm to an object-oriented paradigm is both novel and interesting.
2. As shown in Figure 2, OBJ2CAD is capable of generating complex CAD models with visually appealing results.
3. The figures and tables are self-contained and clearly presented.

**Weaknesses:**

1. The paper lacks analysis of failure cases where OBJ2CAD fails to follow the given instructions.
2. In Figure 1, the meaning of each node in the graph is unclear, and specific examples are missing.
3. The cost associated with using closed-source models in the FEEDBACK MECHANISM is not discussed.

**Questions:**

Please address the weaknesses above.

---

> ### Author Response · Authors · 2025-11-26
>
> Thank you for your valuable feedback. We'd like to address your concerns as follows.
>
> **Q1: lack of failure case analysis**
>
> Appendix B (Demo Details) already includes a representative failure trace from the demo, but for clarity we summarize the major failure modes observed in our broader experiments below.
>
> - **Missing geometric details due to incomplete LLM expansion**. In the demo example, the system first produces a complete and semantically correct high-level plan including "four mounting holes". However, in the first round of shape generation, the LLM instantiates only the base plate and the central adapter, omitting all four mounting holes. This type of failure occurs when the LLM under-expands object definitions or fails to instantiate some sub-objects. In our framework, such errors are typically corrected through iterative feedback and localized regeneration as shown in the demo, where the second iteration correctly adds the holes.
>
> - **More common failures arise from limited spatial reasoning**. For more complex parts, OBJ2CAD failures more frequently stem from incorrect spatial relationships between sub-objects. Typical issues include: inaccurate inference of relative positions, misalignment or unintended gaps, incorrect parameters for arrays, rotational features, or tooth profiles, and mistakes in enforcing concentric, symmetry, or alignment constraints. Although our math conversion mechanism improves geometric reasoning, multi-part assemblies with strong spatial constraints can still exceed the LLM's reasoning capacity.
>
> We have expanded the clarification of these failure cases in Appendix B of the revised paper and refer the reviewer to that section for a complete example.
>
> **Q2: Clarification of graph node in Figure 1 and missing specific examples**
>
> The node types of the object graph are already defined in the caption of Figure 1, formally specified in Definition 2 (Object-Oriented CAD Graph), and further detailed in Appendix A (Graph Representation). For convenience, we provide a clear summary here.
>
> - **Node types in Figure 1(b)**. The object graph in Figure 1(b) directly follows the object graph schema in Figure 1(a), which is derived from the object-oriented FreeCAD code structure. Each node corresponds to one of the schema-defined elements: (i) an object node (a reusable CAD sub-component), (ii) a keyword node, or (iii) a utility function node.
>
> - **Specific example in Figure 2(c)**. The right-hand side of Figure 2(c) presents a real object-level hierarchy generated by our system, including object names, geometric previews, and parent-child relationships, which serves as a concrete instantiation of the object graph. To avoid visual clutter and maintain readability, this example displays only object nodes, intentionally omitting the keyword nodes, utility function nodes, and the semantic edges connecting them. These additional elements are included in Figure 1(b), which illustrates the full multi-type graph structure used for hierarchical semantic retrieval.
>
> We hope this clarification resolves the reviewer's concern.
>
> **Q3: Discussion of the cost associated with using closed-source models**
>
> We thank the reviewer for raising this point. We conduct a detailed cost analysis over the
> 100 samples used in Experiment 1 (Section 4.2), covering both full-generation and feedback stages.
>
> - **Cost of generating a complete CAD part**. Across the 100 samples, the average token usage per sample is 35,156.12 input tokens and 32,854.64 output tokens. Using GPT-5-mini, this corresponds to approximately `$0.074` per sample, and even with GPT-5, the cost remains modest at `$0.37` per sample. Thus, the end-to-end generation is inexpensive and scales well.
>
> - **Cost of the feedback mechanism**. During feedback iterations, we only submit the high-level plan, the relevant object code, and the feedback message to the model. Across our experiments, each feedback round uses roughly 10K input tokens and 5K output tokens on average. This results in a cost on the order of a few cents per iteration with GPT-5-mini, and still well below the cost of a full regeneration pass even when using GPT-5.
>
> Since both the full-generation and iterative feedback stages operate with moderate token budgets,
> the use of closed-source LLMs in OBJ2CAD is cost-effective and operationally feasible.
>
> We have added this cost discussion in Appendix D of the revised paper.

---

### Official Review · Reviewer_DFgT · 2025-11-01

**Soundness:** 2
**Presentation:** 2
**Contribution:** 2
**Rating:** 2
**Confidence:** 3

**Summary:**

The authors propose OBJ2CAD framework that models LLM-based text-to-CAD problem as an object-oriented problem. They also curate a dataset of 1000 examples to align CAD synthesis with human design intuition and propose a generation pipeline with hierarchical retrieval, top-down planning, bottom-up generation, and iterative feedback. The results are showcased on 4 different LLMs on a 100-example test set.

**Strengths:**

- the limitations of procedural CAD generation are well documented
- dataset curation is explained in a sensible manner
- FreeCAD based GUI is practical

**Weaknesses:**

- the claimed shift in paradigm is not substantiated by new representational theory or measurable gains
- no examples of generated designs beyond screenshots are analyzed qualitatively or quantitatively
- "vision consistency" and "human consistency" metrics seem high-level and somewhat subjective
- quantitative metrics (e.g. chamfer distance) or comparisons against procedural baselines are missing
- no measurement of functional correctness or constraint satisfaction

**Questions:**

- I maybe missing something but what is the evidence that "object-oriented paradigm" is actually reducing errors or improving reliability?
- can you run Text2CAD or other methods on your test set and compare success rates, iteration counts, and quality metrics?
- how do you validate functional correctness? because visual similarity does not guarantee gears meshing correctly or holes being concentric or dimensions meeting specifications

---

> ### Author Response · Authors · 2025-11-26
>
> We appreciate the reviewer’s valuable feedback and constructive suggestions. We address the questions and concerns as follows.
>
> **Q1: evidence that object-oriented paradigm reduces errors or improves reliability**
>
> To evaluate if the object-oriented (OO) representation provides measurable benefits over procedural generation, we conduct a controlled experiment on the same set of 100 CAD prompts. We compare four systems, including
> - Procedural-Direct: single-pass CadQuery script
> - Procedural-Decomposed: high-level task plan with CadQuery operations
> - OO-ZeroShot: FreeCAD OO structures without retrieval
> - Full Framework: our complete OO framework
>
> All baselines disable retrieval to isolate the effect of the representation.
> | **Method** | **Compilation Success** | **Vision Consistency** | **Human Consistency** |
> | - | - | - | - |
> | Procedural-Direct | 83% | 35%| 30%|
> | Procedural-Decomposed | 79%| 48%| 45%|
> | OO-ZeroShot | 93%| 53%|50%|
> | Full Framework| 100%| 81%|80%|
>
> These results show both OO-ZeroShot and our full framework achieve more reliable and coherent generation than procedural baselines. This controlled setting further demonstrates that the gains arise from the representational advantage of OO paradigm instead of retrieval or prompt engineering.
>
> Procedural pipelines rely on fragile low-level geometric references and strict operation ordering, often leading to broken constraints, Boolean failures, or cascading collapse from a single mistake. In contrast, FreeCAD's object hierarchy isolates geometry into independent semantic units with explicit attributes and parent-child relations, greatly improving robustness. Also, OO structures align better with human design semantics and naturally support retrieval, reuse and compositional synthesis. These properties enable our full framework to achieve the highest vision and human consistency scores with fewer constraint conflicts which procedural approaches are structurally unable to match.
>
> We have added this controlled comparison in Section 4.3 of the revised paper.
>
> **Q2: comparisons with Text2CAD**
>
> We perform a qualitative comparison between our framework OBJ2CAD and Text2CAD (and several similar open-source procedural CAD systems). However, these models are trained on small operation-sequence corpora using small open-source LLMs, limiting them to produce extremely simple geometries and unable to handle the complexity of our benchmark, let alone industrial CAD tasks.
>
> As shown in Figure 2(a) and Figure 8, when asked to generate several common industrial components, such as hexagonal nut, hex bolt, involute gear, Text2CAD consistently outputs overly simplified shapes with major structural omissions. In contrast, our system reliably produces full-featured and parametrically defined geometries. The discrepancy is not something that can be resolved by prompt engineering; it reflects fundamental differences in representational capacity, model scale, and geometric reasoning.
>
> Because Text2CAD fails to produce valid and compilable geometry for most test cases, it cannot be evaluated under our quantitative metrics. It does not generate enough valid outputs for any meaningful quantitative comparison.
>
> Importantly, even zero-shot generation from a commercial LLM already surpasses Text2CAD by a substantial margin, and our OO framework improves structural coherence even further. For these reasons, we exclude Text2CAD from the main quantitative evaluation.
>
> We have added this qualitative study and visual examples in Section 4.6 and Appendix C.
>
> **Q3: generated designs beyond screenshots**
>
> We have added additional qualitative analyses of generated designs in Section 4.6 and Appendix C, including comparisons with Text2CAD on multiple industrial components.
>
> **Q4: validate functional correctness**
>
> We fully agree that visual similarity alone does not guarantee functional correctness. Our evaluation focuses on geometric and structural correctness, and we do not claim full mechanical validation in this submission. However, our OO formulation provides a significantly stronger basis for functional correctness than procedural generation. In the OO representation, the LLM generates explicit parametric objects, and these parameters and semantic units are available during the planning stage, allowing the model to reason about functional relationships before geometry is instantiated. In contrast, procedural pipelines express designs only as long sequences of low-level operations without explicit semantics or parametric interfaces. Functional correctness appears only at the end of the chain and easily broken, which makes procedural methods unsuitable for validating functional correctness.
>
> Functional validation is an important but separate future stage that must build on reliable structural correctness. Our work establishes this foundation by achieving stable geometric and semantic generation that procedural systems like Text2CAD or DeepCAD cannot support.

---

> ### Author Response · Authors · 2025-12-03
> **Summary Response and Request for Reconsideration**
>
> The reviewer's concerns fall into four main categories: (1) lack of evidence that the object-oriented paradigm improves reliability, (2) absence of comparisons against procedural systems such as Text2CAD, (3) no qualitative or quantitative analysis beyond demo screenshots, and (4) insufficient discussion of functional correctness. We have substantially revised the paper to directly address all of these points:
>
> - **New Section 4.3 now includes a new controlled experiment** comparing two procedural baselines (Procedural-Direct, Procedural-Decomposed) with two OO variants (OO-ZeroShot, our proposed OBJ2CAD). These results show clear, measurable reliability gains attributable directly to the OO paradigm.
>
> - **New Section 4.6 and Appendix C now include expanded qualitative comparisons with Text2CAD and visualized examples** for several industrial components (hex nut, bolt, involute gear). These results illustrate that OBJ2CAD produces full-featured and parametrically defined geometries while procedural systems consistently generate incomplete or invalid shapes, and that Text2CAD does not produce enough valid results for quantitative evaluation.
>
> - **Section 4.6 and Appendix C now include qualitative analyses of generated designs, and Appendix B includes failure-case studies**, addressing the request for analysis beyond screenshots.
>
> - Functional correctness is now explicitly discussed as a separate stage that requires semantically structured geometry. We argue that OO representations provide the necessary foundation, whereas procedural representations fundamentally cannot support functional validation.
>
> We would also like to highlight the pioneering nature and significance of this work. Object-oriented text-to-CAD synthesis is an underexplored direction with substantial potential for real-world industrial applications. No prior work has introduced an OO formulation, an OO dataset, or a framework enabling hierarchical retrieval, top-down planning, and object-level assembly reasoning. Our dataset, methodology, and analysis establish the first foundation for this emerging research direction. Given the added experiments, additional analyses, and clarified explanations, **we hope the reviewer can reconsider the rating** to support further exploration and innovation in this domain.

---

### Note · Program_Chairs · 2026-01-17
**Submission Desk Rejected by Program Chairs**

The following references in this submission do not refer to real documents and/or have major errors in bibliographic information:

 Zezhou Zhang, Xinhai Chen, Erich Gan, Karl D. D. Willis, and Wojciech Matusik. Cadtransformer: A hierarchical transformer for computer-aided design sketches. In Proceedings of the IEEE/CVF International Conference on Computer Vision (ICCV), pp. 12589-12599, 2021.
Xinyu Xu et al. Llms and the abstraction and reasoning corpus: Successes, failures, and the importance of object-based representations. arXiv preprint arXiv:2305.18354, 2023b.
Xinyu Xu, Karl D. D. Willis, Boxi Du, Erich Gan, and Wojciech Matusik. Auto-regressive cad modeling with transformers. arXiv preprint arXiv:2301.13163, 2023a.